# Comparative Modeling and Analysis of Extremophilic D-Ala-D-Ala Carboxypeptidases

**DOI:** 10.3390/biom13020328

**Published:** 2023-02-09

**Authors:** Elizabeth M. Diessner, Gemma R. Takahashi, Rachel W. Martin, Carter T. Butts

**Affiliations:** 1Department of Chemistry, University of California, Irvine, CA 92697, USA; 2Department of Molecular Biology and Biochemistry, University of California, Irvine, CA 92697, USA; 3Departments of Sociology, Statistics, Electrical Engineering and Computer Science, University of California, Irvine, CA 92697, USA

**Keywords:** serine protease, MEROPS S11, comparative modeling, molecular dynamics, extremophiles, machine learning

## Abstract

Understanding the molecular adaptations of organisms to extreme environments requires a comparative analysis of protein structure, function, and dynamics across species found in different environmental conditions. Computational studies can be particularly useful in this pursuit, allowing exploratory studies of large numbers of proteins under different thermal and chemical conditions that would be infeasible to carry out experimentally. Here, we perform such a study of the MEROPS family S11, S12, and S13 proteases from psychophilic, mesophilic, and thermophilic bacteria. Using a combination of protein structure prediction, atomistic molecular dynamics, and trajectory analysis, we examine both conserved features and trends across thermal groups. Our findings suggest a number of hypotheses for experimental investigation.

## 1. Introduction

S11 proteases are specialized carboxypeptidases that catalyze the cleavage of D-Ala-D-Ala peptide bonds during bacterial cell wall biosynthesis, limiting the length of the polypeptide stems that form cross-links between glycan chains in the sacculus [1]. This activity plays a critical role in maintaining a normal cell shape [2,3]. Although many organisms appear to have multiple redundant S11 carboxypeptidase paralogs, specific paralogs are needed for growth under challenging environmental conditions, e.g., metal ion-depletion [4,5]. Organisms such as *E. coli* that grow under a wide range of conditions have a portfolio of different carboxypeptideases that are active over different pH ranges [6,7].

S11 carboxypeptidases are also called penicillin-binding proteins because they bind and are inhibited by penicillin and related β-lactam antibiotics due to the structural similarity between these molecules and their dipeptide substrate or the transition state of the reaction they catalyze [8,9,10]. The catalytic triad consists of two serine residues and one lysine embedded in the strongly conserved motifs SxxK and SxN, where x is variable [11]. A second lysine, which is part of the motif KTG, is highly conserved and may participate in polarizing the catalytic serine and thus increasing its nucleophilicity [12,13]. In the normal S11 protease catalytic cycle, the nucleophilic serine attacks the D-Ala carbonyl group of the peptide bond between the two D-Ala residues, making a covalent acyl intermediate. A water molecule then attacks the acyl intermediate, leading to the release of the product, in this case, a free D-Ala and a shortened peptide stem on the cell wall. Inhibition by β-lactams works by slowing down the release of the product, effectively leaving the active site permanently blocked via covalent modification. Although S12 and S13 carboxypeptidases differ somewhat in their overall molecular architecture, they share the SxxK active motif, a similar catalytic domain fold, and a common enzymatic mechanism with the S11 proteases; therefore, they are also included in this study.

As essential enzymes for bacterial growth, the S11, S12, and S13 proteases are present not only in mesophilic bacteria but also those growing in extreme conditions. As such, they provide natural targets for comparative analysis. In this paper, we employ structure prediction, molecular modeling, and analysis to investigate multiple proteases of these families from a selection of psychrophilic, mesophilic, and thermophilic bacteria. We model the proteases under native thermal, pH, and ionic concentration conditions, allowing a comparison of predicted properties that take into account the environment in which bacterial growth occurs. Our findings suggest a number of hypotheses regarding both conserved and differentiating characteristics across thermal groups, which are potentially fruitful targets for experimental research. In particular, we find that thermophilic proteins in this group are primarily stabilized by salt bridges, not disulfide bonds or enhanced hydrophobic packing, while psychrophilic proteins have a more diverse range of adaptations.

## 2. Materials and Methods

### 2.1. Sequence Selection, Alignment, and Clustering

Eleven organisms from different thermal environments (psychrophile, mesophile, and thermophile) were initially selected for this study based on the estimated thermal conditions of their documented habitats as well as descriptions in the literature (see below). Serine proteases were then identified in the UniProt proteomes from these organisms. The descriptions and environments of the selected strains (from which these proteomes were annotated) largely match their respective thermal categories. Two proteomes from *Shewanella frigidimarina* were used because one (SHEFN) was derived from a presumably less psychrophilic isolate in the North Sea near Aberdeen, UK, whereas the other (SHEFR) was from an Antarctic isolate (see SHEFN and SHEFR below).

The following proteomes were chosen based on the above criteria: five psychrophilic (*Pseudoalteromonas translucida* (ID PSET1, UniProt UP000006843, RefSeq GCF_000026085.1) [14], *Psychrobacter arcticus* (ID PSYA2, UniProt UP000000546, RefSeq GCF_000012305.1) [15], *Psychromonas ingrahamii* (ID PSYIN, UniProt UP000000639, RefSeq GCF_000015285.1) [16], *Shewanella frigidimarina* (ID SHEFN, UniProt UP000000684, RefSeq GCF_000014705.1) [17], *Shewanella frigidimarina* (ID SHEFR, UniProt UP000055702, RefSeq GCF_001529365.1) [18]); three mesophilic (*Bacillus subtilis* (ID BACSU, UniProt UP000001570, RefSeq GCF_000009045.1) [19], *Escherichia coli* (ID ECO57, UniProt UP000000558, RefSeq GCF_000008865.2) [20,21], *Pseudomonas fluorescens* (ID PSEPF, UniProt UP000002704, RefSeq GCF_000012445.1) [22]); and three thermophilic (*Caldicellulosiruptor saccharolyticus* (ID CALS8, UniProt UP000000256, RefSeq GCF_000016545.1) [23,24], *Geobacillus kaustophilus* (ID GEOKA, UniProt UP000001172, RefSeq GCF_000009785.1) [25], *Thermomicrobium roseum* (ID THERP, UniProt UP000000447, RefSeq GCF_000021685.1) [26]. Metadata about the environment, including observed environmental temperature and pH, were obtained from the NCBI entries for each organism and associated publications. We use the observed environmental temperature recorded for each isolate (Appendix A) in assigning proteins to thermal groups and in determining biologically relevant modeling conditions. We note, in passing, that use of the observed environmental temperature avoids the numerous complications involved in attempting to define and identify an optimal growth temperature for each organism, a quantity that can depend on the growth criterion used, available media, other environmental conditions and details of colony selection during the experiment. Further, optimal growth conditions for the organism, and may be imperfectly related to those for the enzyme itself. Since the temperatures used here correspond to experimental observations of the organisms growing in the wild (and to the specific isolates used as our source of sequence information), the enzymes studied here are functional at the specified temperatures. We do not, however, make any claims about the optimality of these conditions for either enzyme performance or bacterial growth. Measured intracellular pH values and salt concentrations were used in cases where they have been experimentally measured and reported in NCBI. Otherwise, pH 7.2 and 200 mM salt were assumed.

Proteins with putative serine proteolytic activity (henceforth referred to as serine proteases) were curated from these proteomes using a set of predetermined criteria. UniProt [27] proteomes were queried for all [[GO: serine-type] AND [NOT GO: inhibitor]] proteins. Additional proteins that were not retained in the initial query were manually screened for putative proteolytic activity if they satisfied any of the following criteria: [[GO: peptidase activity [0008233]] AND [Protein Name: serine]] or [[Protein Name: serine] AND [Protein Name: protease]] or [[Protein Name: serine] AND [Protein Name: peptidase]] or [[Protein Name: serine] AND [Protein Name: proteinase]]. Proteins that satisfied the above criteria were flagged if they contained transmembrane regions and/or intramembrane regions, identified in either UniProt or Scampi2 [28]; the remaining unflagged proteins were retained without the need for further processing. Flagged proteins whose predicted membrane regions lay within the first 70 amino acids (inclusive) were unflagged if they also contained a signal sequence predicted by SignalP-6.1 [29]. Finally, two Clp protease ATP-binding subunits (ClpA) were removed from the dataset by hand, as they do not have proteolytic activity. At this point, the remaining unflagged proteins were considered to be putative non-membrane serine proteases and were compiled for further analysis. All sequences from the study organisms that were annotated as coding for MEROPS S11 serine proteases using a BLAST search against SwissProt and InterProScan [30] were selected, resulting in 24 S11 proteases sequences in the final data set: 7 from thermophiles, 9 from mesophiles, and 8 from psychrophiles. One S12 protease from *E. coli* was found, along with seven S13 proteases, three from mesophiles, and four from psychrophiles.

Ward’s method (in the generalization of [31]) was employed to construct a hierarchical clustering of S11 serine proteases based on sequence dissimilarity. ClustalOmega was used to produce sequence alignments [32]. The following settings were used: gap open penalty = 10.0, gap extension penalty = 0.05, hydrophilic residues = GPSNDQERK, and a BLOSUM weight matrix. Logo plots showing conservation of sequence regions were generated using WebLogo 3.0 [33]

### 2.2. Structure Prediction

The initial structures were predicted using the iTasser software pipeline [34,35], which uses comparative modeling and all-atom refinement based on a simplified forcefield. Models were also generated using AlphaFold [36] but were found to be unsatisfactory for most proteins in this data set, with long unstructured regions that are not compatible with other information about these proteins. We discuss this issue further in Section 4.1. Using the iTasser structures as a starting point, we then performed in silico maturation on these preliminary models, where signal sequences (identified using SignalP 6 [29]) and pro-sequences were removed, and the protonation states at the relevant pH for each organism were adjusted using PROPKA3 [37]. Finally, each modified structure was equilibrated in explicit solvent using atomistic molecular dynamics, as described below. The PDB files corresponding to the in silico matured structures are available in the Appendix A. Protein structure figures were generated using UCSF Chimera [38] and VMD [39].

### 2.3. Molecular Modeling and Analysis

After the removal of signal sequences and pro-sequences, structures of the proteases described above were modeled in explicit (TIP3P [40]) solvent using atomistic molecular dynamics (MD) simulations. The correction of protonation states for native pH was performed using PROPKA3 [37], and the systems were prepared in cubic water boxes with a 15 Å minimum margin. Counterions were added to neutralize the structure, and additional ions (NaCl) were added to match the system ionic strength; where this was unknown, 200 mM salt was employed (All structure preparation was performed using VMD [39]). The prepared system was then equilibrated as follows: All simulations were performed in NAMD [41] using the CHARMM36m force field [42,43] under periodic boundary conditions in an NpT ensemble at 1 atm pressure with Nosé–Hoover Langevin piston pressure control [44,45] and Langevin temperature control (damping coefficient 1/ps). To obtain representative structures for each protein, they were simulated for 1 ns at the observed environmental temperature for each organism (following a short (10 ps) minimization and box size adjustment period), with the final frame being employed for visualization and analysis. Long trajectories were also obtained for selected systems. These systems were prepared as above but were simulated for 100 ns, with 5000 frames retained for analysis (1/20 ps) (RMSD and energy plots are shown in Appendix A).

All frames (both single structure and 100 ns trajectory cases) were analyzed using DSSP [46] to obtain residue exposure and secondary structure information. Surface areas of the residues were obtained from Pacios [47], and were used to calculated the total surface area of each sequence. The solvent accessible surface area (SASA) values obtained from DSSP output were used in combination with the total surface area to calculate a packing metric for comparison of protease structures based on the internal and surface residue interactions. The packing metric was determined by taking the total surface area of each sequence, that is, the sum of the total surface area of each residue minus the total sum of SASA values for that sequence. This gives an approximation of the amount of surface area within the protease structure that is not interacting with the solvent and, therefore, must be interactions between residues of the same sequence. The buried surface area was then divided by the total SASA for each sequence to give a ratio that serves as a rough packing metric.

SASA values were also used to visualize the fluctuations in surface residues on protease structures. The range of each residue RSA was calculated as the maximum SASA value minus the minimum SASA value, then divided by the residue’s total surface area. The hydrophobicity scale of Kyte and Doolittle [48] for amino acids was used for the comparison of exposure and burial rates of individual residues among structures of each sequence cluster. An individual residue is considered “exposed” when its RSA is greater than 20%.

Finally, each single-frame model was represented as a protein structure network (PSN), using the node definitions of Benson and Daggett [49], in which each vertex corresponds to a single moiety within the protein. We follow the network construction procedure of Butts et al. [50], in which two vertices are adjacent within the PSN if their corresponding moieties have at least one atom pair in contact (here defined as being within 1.1 times the sum of their van der Waals radii). Degree (number of contacts) and degree *k*-cores (suggested as a measure of local cohesion in PSNs by [51]) were computed for all PSNs and retained for subsequent analysis. Network construction and analysis were performed using the sna [52] and network [53] libraries for R [54].

### 2.4. Random Feature Clustering and Markov Analysis

To compare active site dynamics, we perform an analysis of the dynamics of the three catalytic residues identified by MEROPS for S11, S12, and S13 families (the primary serine and lysine, and the second stabilizing serine) using a combination of random feature projection, cluster analysis, and Bayesian Markov modeling. For each simulated trajectory, we first extract the coordinates of all heavy atoms associated with the catalytic residues (using the bio3d library for R [46,55]) in each frame, converting these to frame-wise interatomic distance matrices. These per-frame matrices were vectorized using the sna library for R [52] to create a frame by distance matrix for each trajectory. These were, in turn, concatenated and then projected onto a deep random feature space based on a stacked composite linear and arc-cosine kernel model [56]. The procedure employed was as follows: first, the input distance vectors were augmented with a constant (intercept) feature and projected onto 500 random ReLU functions (unit radius spherical projection). These 500 projected values were then augmented with a copy of the inputs (i.e., a skip layer), and pruned to 25 features by principal component analysis (PCA). The resulting PCA scores were then used as inputs additional layers formed in the same fashion; in total, six layers were employed, with the 25 PCA features from the final layer used as the data embedding (Embedding was implemented in R; portions were written using the Rcpp package [57]). A scree plot on the final extracted dimensions (see Appendix A) shows that 25 dimensions are sufficient to account for observed dynamics. We note that all trajectories were projected jointly into the same space, allowing for direct comparison both within and between trajectories.

Given the 25-dimensional joint embedding, conformations were clustered into conformational states using *k*-means clustering (default R implementation, 75 restarts, 100 iterations). The number of clusters was determined using the total Markov error method of [58], with transition rates and associated errors estimated within-trajectory; *k* was chosen to be the largest number of clusters having a posterior predictive *Z*-score for the reduction in mean RMSE over trajectories greater than 2 (see Appendix A) (RMSE computed using pairwise interatomic distances on the original scale). This resulted in five clusters. Posterior mean occupancy rates and transition rates were obtained for each cluster under Jeffreys priors, per [58].

To obtain further insight into the key determinants of cluster membership on the original distance scale, a classification tree was used to predict cluster membership from raw conformations, expressed as the original vectors of heavy atom distances. Tree fitting was performed using rpart [59], with a maximum depth of 3, cross-validation sample size of 150, and complexity parameter of 0.001. The distances and distance thresholds found to separate the conformational states were then used for the interpretation of the clusters, as discussed below.

## 3. Results

### 3.1. Sequence Analysis Shows Three Temperature-Related S11 Protease Groups

Sequences of S11 serine proteases were clustered by sequence similarity using multidimensional scaling (Appendix A) and Ward’s algorithm as implemented in R [54], resulting in three main clusters (Figure 1). In the dendrogram, the first split separates out a small group of five psychrophilic and mesophilic sequences, and the second split divides the remaining sequences into thermophiles and psychrophiles, with some mesophiles in each group. The corresponding sequence alignments are shown in the SI (Appendix A). All of the proteases from the mesophilic organism *Bacillus subtilis* (BACSU) were clustered with the thermophilic enzymes, while the mesophilic proteases from *Escherichia coli* (ECO57) and *Pseudomonas fluorescens* (PSEPF) were distributed among the two psychrophilic protease groups. Figure 1 shows the dendrogram for the clustered sequences, with the thermophilic group (A) shown in red and the psychrophilic groups (B) and (C) shown in light and dark blue, respectively. The centroids of each cluster, the sequences that are the most similar to all other sequences in the same cluster, are highlighted in the color of their respective groups. The active site residues, including the catalytic triad and an additional conserved lysine that is believed to be important for substrate binding and stabilization of transition states, are located in three sequence blocks that are characteristic of their respective clusters. Sequence logos illustrating the distribution of residues in these blocks for each cluster are shown in Figure 1D–F. The annotated sequence of the one S12 protease in this data set is shown in Appendix A, while the sequence dendrogram and alignments for the S13 proteases are shown in Appendix A, respectively. The sequence diversity of these clusters varies considerably, with relatively few strictly conserved residues away from the active site in S11 cluster A and S13 and more widespread sequence conservation in S11 clusters B and C.

### 3.2. Thermophilic Enzymes Are Enriched in Charged Amino Acids, but Not Hydrophobic Ones, and Psychrophilic Enzyme Composition Differs Markedly by Group

To investigate the hypothesis that proteins adapt to higher temperatures by increasing the number or sizes of hydrophobic amino acids [60], an analysis of the primary structure was performed in which counts of each amino acid were divided by the total sequence length to give a relative amino acid composition of each sequence. The results (Figure 2) show an increase in amino acids K, E, Y, and V with increasing temperature. In contrast to this clear trend for thermophiles, many amino acids are increased in one of the psychrophilic groups relative to the thermophilic group but decreased in the other psychrophilic group. Amino acids R, T, A, and L are more strongly represented in psychrophilic group C, while D, W, G, F, and I are more strongly represented in psychrophilic group B. Amino acids that increase in abundance in both psychrophilic groups are N, Q, H, P, S, and M. Contra the conventional wisdom that thermophilic proteins are often stabilized by disulfide bonds (which other groups have also pointed out is true for some enzyme classes but not others [61]), cysteine is less common than average in this set of proteins, regardless of the thermal group. Of the enzymes observed here, disulfide bonds are only observed in the S13 proteases, all of which are meso- or psychrophilic. Therefore, while disulfide bonds are known to be stabilizing structural components in some thermophilic proteins [62], that mechanism is not at play for these D-Ala-D-Ala carboxypeptidases.

Figure 3 shows the amino acid composition grouped by type. Here, we observe a greater concentration of uncharged polar residues in psychrophiles. Both positively and negatively charged residues (R, K, D, and E) have a higher abundance in thermophiles, potentially allowing for more stabilizing salt bridges; we investigate this possibility below.

Thermophiles were observed to contain larger numbers of charged amino acids (at the expense of uncharged polar residues) than their psychrophilic homologs, and valine was the only hydrophobic residue observed to have increased prevalence with temperature. Both patterns have previously been observed in other systems by Kumar and Nussinov [63]. Another pattern hypothesized to be present, an increase in proline for enzymes adapted to higher temperatures, was not observed. However, increased proline composition has been argued to stabilize thermophilic enzymes by decreasing the entropy of the unfolded state [64], here that adaptation does not appear to be necessary. It may be worthwhile to further explore how proline placement affects conformational and functional dynamics in these and other thermophilic proteins. Among the psychrophiles, it is notable that compositional variation between sequence groups is often similar to or greater than the differences between either group and the thermophiles. This suggests that adaptation to the thermal environment may place fewer constraints on amino acid composition than other factors.

Locally equilibrated structures of the S11 protease sequence cluster centers are shown in Figure 4A–C. All the proteins share a common fold for the catalytic region, an α-β-α sandwich with the active site between Domains 1 and 2. The proteins in clusters A and B also have a C-terminal tail domain that is believed to be important for anchoring these proteases to the cell wall as it is being synthesized [65]. This domain is primarily composed of β-strands, although many have a short helix at the C-terminus, which protrudes from the tail domain and may be important for interaction with the growing cell wall [66]. The insets show close-up views of the active residues, including the catalytic triad and additional functional lysine residue for each enzyme. The nucleophilic serine and the lysine that serves as the general base are located in a conserved SxxK motif, where x can be any amino acid [67]. Here, the first residue is always L in cluster B and varies in clusters A and C. The second residue is always T in clusters A and B, and is T or S in cluster C. The second serine resides in an SxN motif, where x is evenly split between A and G in clusters A and B and is always E in cluster C. The second lysine, which is part of a conserved KTG sequence, is positioned on the β-strand nearest the active site and appears to interact with the backbone of the α-helix containing the SxxK residue motif, potentially participating in substrate binding and/or increasing reactivity of the main catalytic residues [68]. Example structures for the S12 and S13 proteases are shown in Appendix A.

#### Composition by Exposure

We examine the distribution of exposure and burial rates of all residues of each amino acid type for all proteases in Figure 5. “Exposure” on an individual residue level was defined to occur when the DSSP calculation of SASA for a single residue was more than 20% of the total surface area for that residue. All exposed residues of an amino acid type were divided by the total number of exposed residues, giving the composition of the protein surface. A similar calculation was performed for buried residues, but only including residues that had less than 20% of the total surface area exposed to the solvent.

The broad range of surface composition for the hydrophilic R and K residues in thermophiles reflects the variation in the portion of the total sequence held by those residues (Figure 2) and suggests a role in mediating solvent and substrate interactions on the surface. In thermophiles, the overall sequence enrichment in E causes it to be more abundant both at the surface and in the interior. Thermophiles also have a lower abundance of N, Q, P, A, and F at the surface. All other amino acids, including the more hydrophobic L, V, and I, occupy the same portion of the surface across thermal groups. In the interior, thermophiles are observed to be enriched in A and exiguous in N, S, and M, with all other amino acids having similar relative abundance across groups.

### 3.3. Packing and Flexibility Favor Function and Interaction over Stability at Extreme Temperatures

Protein structure is known to be affected by temperature, as higher temperatures increase the magnitude and frequency of molecular motion. This intuitively leads to the conjecture that a protein must adapt to higher temperatures by increasing the density and rigidity of its structure to withstand fluctuations and prevent unfolding. We hypothesized that this set of proteins would also show increases in density and rigidity with increasing temperature. Additional molecular dynamics simulations and protein structure network analysis were performed to observe differences in the related measures of packing and flexibility. Using the iTasser models with signal sequences removed as a starting point, proteins were simulated at their observed environmental temperatures (Appendix A), and the final frame of a 1 ns simulation was used for the analysis of each S11, S12, and S13 protease. As part of these analyses, the solvent accessible surface area (SASA) of each structure was calculated using the DSSP [69] function as part of the bio3d [46,55] package in R.

#### 3.3.1. Packing

For a structure-level comparison of residue interactions of different proteins, a packing metric was created by determining the ratio of buried to exposed surface area in the tertiary structure of a protein. To find the buried surface area, the total surface area of each protein was calculated using residue surface areas from Pacios [47] summed for the entire sequence minus the SASA values computed with DSSP. The resulting surface area value was then divided by the total sum of SASA values for the sequence to give a ratio of buried to exposed surface area. The ratios for each protein are shown in Figure 6, with the proteases ordered according to their position in the cluster dendrogram of Figure 1. The resulting plot shows a qualitative trend of increased packing with decreased temperature; proteases in the thermophile group, in general, have less internal (i.e., buried) surface area relative to SASA, while those in the two psychrophile groups have more relative internal surface area. The amount of internal surface area intuitively corresponds with residue interactions within the protein, with an increase in the relative amount of internal interactions being used here as a rough proxy for the packing and stability of a protease.

The relationships shown above were obtained from equilibrated snapshots (i.e., single structural predictions) and may be an imperfect depiction of long-run behavior. To test whether the thermal trends seen in Figure 6 hold on longer time scales, a sample of proteases was simulated for 100 ns at their observed environmental temperatures. The dynamic packing ratio (buried vs. exposed surface area) was again plotted for the simulated trajectories as the autocorrelated mean over 5000 frames, shown in Figure 7 (95% confidence intervals for each point were calculated but are too narrow to be visible). The trajectories show a similar qualitative trend in packing versus thermal group, although the values of the ratios are lower in some cases. The lower value is reasonable given that there can be assumed to be a limit at which the protein can no longer be compressed, limiting the total amount of buried surface area. On the other hand, the limit for expanding the protein is complete unfolding, which does not occur in these simulations. Because of the nature of these limits, the simulated protein would be more likely to experience transient conformations where there is a greater amount of exposed surface area, leading to a lower mean ratio of buried versus exposed surface area. Along those lines, the thermophilic proteases experience a notable decrease in the packing ratio, while the one mesophilic protease in the thermophile group shows very little change. For the psychrophilic groups, there is a slight decrease in packing ratios, similar in magnitude to that of the thermophilic group. Despite these changes, however, the overall trend remains, where the majority of thermophilic proteases have a lower ratio of buried to exposed surface area than their psychrophilic counterparts.

The notable exception in the thermophilic group, B9KYG0_THERP, has a very high packing ratio, on par with the S12 and S13 serine proteases. This, as we show below, appears to stem from the fact that Domain 3, the “tail” domain of B9KYG0_THERP, is folded onto the main structure, resulting in a highly compact protein with a large core. The reason for this distinctive structural feature is not known and would appear to be an interesting target for experimental investigation.

Given the observation of diminished packing with temperature, it is reasonable to ask whether surface stability follows a similar pattern. In addition to the potential selection of thermophilic enzymes for greater stability, the enhanced strength of the hydrophobic effect at higher temperatures could be expected to lead to a stronger separation between exposed hydrophilic and buried hydrophobic residues, leading to greater consistency in solvent exposure than in psychrophiles. Residue interactions with solvent were examined by calculating the dynamic variation in SASA values for each residue over the course of each simulated trajectory. This was performed by taking the difference in the range of SASA values observed for each residue per trajectory and then dividing by that residue’s total surface area to give a value for the residue’s relative range of exposure (i.e., for residue *i* with SASA value $s_{it}$ at time t∈T and total SA $A_{i}$, the index $D_{i}=\left(\max_{t\in T}s_{it}-\min_{t\in T}s_{it}\right)/A_{i}$). High values of the dynamic solvation index indicate surface residues in flexible regions that see considerable changes in solvation, while low values indicate either stable burial or exposure. The $D_{i}$ values were mapped to the surface of each protease and visualized in VMD. The resulting structures are shown in Figure 8, in which the structures are arranged to replicate the layout of the values plotted in Figure 7 for ease of comparison.

The dynamic solvation index values suggest high levels of surface flexibility in the tail domain (particularly for thermophiles) and for particular residues near the active site. Although some thermophiles have regions with particularly high levels of dynamic solvation, this is not universal (see, e.g., A4XKJ5), and all psychrophiles also show regions of high *D* value. In turn, we see notably low *D* values for large areas of the primary domain in the psychrophilic proteases, as well as isolated patches on the thermophilic and S13 proteases. We also note a high instance of low *D* values on the surface on the opposed (antipodean) side of the protein to the active site, suggesting a high level of surface stability in this region. The disagreement in *D* values across the surface of the proteins indicates uneven solvent interactions, which suggests that the patterns are unlikely to be due to general factors such as differences in the strength of the hydrophobic effect.

Another observation from Figure 8 is the difference in the structure of the outlying thermophile, B9KYG0_THERP (names are shortened on the plot for clarity). As noted above, this thermophilic protease differs from the rest of the thermophiles by its apparent lack of the “tail” domain that extends from the left side of many of the S11 proteases shown in Figure 8. A closer look at the structure reveals that this domain is not, in fact, absent but more closely bound to the rest of the protease. This explains its higher packing ratio, as the protease buries a greater number of residues by folding the “tail” domain against residues that are exposed to the other proteases examined here. Given the high-temperature conditions in which this protease typically operates and was modeled (≈346 K), it is plausible that the integrated tail structure is a stabilizing adaptation. Notably, the domain interface between the “tail” and the main body of the protease includes residues on the far end of the α-helix holding the SxxK catalytic residue motif. On the other hand, the smaller psychrophile group (group C) is observed to actually lack the “tail” domain altogether. If the tail is indeed a binding domain, it is possible that it is not necessary at low temperatures (where interaction with the body of the protease may be sufficient) or that proteases in this group have different binding partners.

#### 3.3.2. Flexibility

Decreased packing density with increased environmental temperature could, in principle, be offset by the increased local density of contacts between chemical groups within the protein (i.e., structural cohesion), providing more paths through which energy obtained by thermal collisions could be diffused without disrupting the protein’s fold. To examine this possibility, we turn to protein structure networks. Both the mean degree (average number of contacts per group) and the mean degree *k*-core number (i.e., the number of the highest degree-core to which a vertex belongs) serve as measures of local cohesion, the first capturing direct contacts within the protein, and the second capturing the extent to which chemical groups are embedded in sub-structures connected by multiple, redundant paths [51]. Figure 9 shows the relationship between both cohesion measures and observed environmental temperature. Contrary to expectation, we observe clear trends towards lower levels of cohesion at higher temperatures (Pearson correlation ρ of −0.61 and −0.55 for mean degree and core number, respectively, both p<0.01), with corresponding trends toward greater cohesion at low temperatures. Further, we observe no thermophilic enzymes with markedly high levels of cohesion and no psychrophiles with very low levels of cohesion; much of the statistical noise in the relationship comes from mesophiles, which are observed to run the gamut from very loose to very tight structures. Many mesophiles also have significant overlap in their range of growth conditions with thermophiles, psychrophiles, or both, i.e., many organisms that are capable of surviving temperature extremes are facultative extremophiles that grow even better under moderate conditions. This could potentially imply that the effect results from constraints on the ability to adaptively *avoid* high cohesion at low temperatures or low cohesion at high temperatures rather than selection for a thermally optimal level of cohesion at each temperature.

Is the cohesion trend driven entirely by differences in structure on the protein surface (or, respectively, in the core)? Figure 10 shows that this is not the case. While nodes associated with buried residues have much higher mean core numbers than exposed residues, higher environmental temperature is negatively correlated with a core number for both sets of nodes. It does not, therefore, appear to be the case that, e.g., thermophilic proteins within this set are preferentially held together by an especially cohesive core, nor that the psychrophilic proteins have evolved a particularly “loose” surface structure to facilitate substrate binding or other conformation-dependent processes.

Because stability is essential to maintaining function at high temperature and flexibility to maintaining function at low temperatures, one might reasonably hypothesize greater packing (more internal interactions) and greater structural cohesion within thermophilic enzymes than psychrophilic ones; taken together, our results clearly contradict this hypothesis. One alternative explanation is that, despite respective adaptations for stability or flexibility, temperature-dependent fluctuations nevertheless result in more average exposure at high temperatures than at low temperatures. The results shown above on solvation dynamics are roughly compatible with this possibility. However, we also observe that, in the systems studied here, the thermophilic enzymes appear to have more elongated and flexible structures than psychrophiles at the domain level, with marked differences in cohesion throughout the protein (examples are shown in Appendix A), and slightly lower proportions of organized secondary structure (Appendix A). This suggests that while the secondary structure is strongly related to local cohesion (moieties associated with helices are generally more cohesively embedded than those in sheets, both of which are much more cohesively embedded than those in other secondary conformations), within-conformation correlation of cohesion with temperature is clear only for chemical groups in the “random coil” (non-helical, non-extended) secondary structure (Appendix A). Thus, at least within the set of proteins studied here, the temperature-cohesion relationship seems to involve a higher-order structure, and does not appear to be entirely a consequence of local interactions in the secondary structure.

#### 3.3.3. Active Site Structure

While our expectations regarding protein adaptation to extreme thermal environments have been largely contradicted by our results, we note that the stability of the entire protein should be a lower priority for adaptation than the functionality of the active site. We hypothesize that, should differences exist between the active sites of thermally diverse homologs, those differences will be temperature dependent. To compare active site conformational dynamics uniformly across proteins, we focus on the three residues considered by MEROPS to be directly involved in catalysis: the primary serine and lysine of block 1 and the secondary serine of block 2. Visualization of the three dimensions with the highest variation in the deep random feature embedding for the trajectory set is shown in Figure 11; RGB color hinting for each dimension facilitates identifying dimensionally separated points in pairs plots. We can clearly see several distinct conformational states, along with one high-occupancy cluster. Minimization of the total Markov error when using clustering for stepwise structure prediction leads to a 5-cluster solution, which we employ below.

To aid in interpreting the conformational clusters, we train a classification tree on cluster membership, with inputs being the original (unembedded) heavy atom distances among the catalytic residues. Figure 12 shows the resulting tree structure, along with partition information for each node. Decisions at each node are made by thresholding a specific distance (expressed in CHARMM notation, with S1 being the principal serine, K being the lysine, and S2 being the secondary serine); thus, e.g., “OG.S2.O.S1 ≥ 8” indicates a decision that thresholds the distance between catalytic oxygen (OG) of serine S2 and the backbone carbonyl oxygen (O) of serine S1 at 8 Å. Seven distances were identified as important for distinguishing among states: S1 carbonyl oxygen vs. S2 catalytic oxygen; K carbonyl carbon vs. S2 proton HG1; S1 catalytic oxygen vs. S2 carbonyl carbon; lysine beta carbon vs. S2 alpha carbon; S1 catalytic oxygen vs. S2 catalytic oxygen; S1 carbonyl oxygen vs. S2 carbonyl carbon; and S1 carbonyl oxygen vs. S2 alpha carbon. We can immediately see from the tree that cluster 3 is the most extended and cluster 4 is the most compact, with cluster 1 differing from cluster 3 largely by the closer placement of the S1 and K backbone atoms. Clusters 2 and 5 are more intermediate, with cluster 5 differing from cluster 2 by having a larger distance between the side-chain oxygens of S1 and S2.

Further insight can be obtained by examining the respective most central conformations observed in each cluster, as shown in Figure 13. Cluster 4 is, as expected from the classification tree result, a compact conformation in which the S1 backbone carbonyl is stabilized by the K side chain, with the S2 side chain close to and parallel to the lysine. This is compatible with a stabilizing role for S2, as also seen in clusters 2 and 5. By contrast, in cluster 1, we see that S2 is oriented away from the rest of the active site, while in cluster 3, the S2 side chain is oriented toward the K backbone (and not the side chain, as in the stabilizing configurations). Interestingly, a “classic” serine hydrolase conformation with the catalytic oxygen of S1 oriented towards the proton source (here, lysine) is only seen in cluster 3, with S2 occupying this more typical position in clusters 2, 3, and 5. This suggests that substrate binding is required to produce typical catalytic dyad (S1-K) conformations in this group of hydrolases or (perhaps more intriguingly) that S2 may, in some cases, be able to fill the role of S1 here.

Insights into how conformations are related to sequence can be seen by an examination of state occupancies and estimated transition rates by protein, as shown in Figure 14. Not every protein needs to occupy every state, and indeed we see some segmentation; most notably, states 3 and 5 appear characteristic of single proteins, A0A119D0G3_SHEFR and A0A0H3JIB3_ECO57, though both proteins also spend some fraction of their time in other states. In turn, a small number of proteins (Q4FTU5_PSYA2 in state 1, and A1SZ14_PSYIN, Q3K9W6_ PSEPF, and A1SZR5_PSYIN in state 4) spend all of their time in one state, though they share their respective states with other proteins.

In general, we thus see some mixing across conformational states, though many proteins are heavily concentrated in particular states. The thermophilic cluster (as well as the S13 group) is observed to mix across more states than either psychrophilic cluster, which is possibly explained by the increased rate of transition between conformations afforded to proteins at higher temperatures, which naturally lends to increased sampling of conformations that otherwise may not be observed on a short time scale of an MD simulation. The popularity of state 4, which is occupied by all proteins but three (A0A119D0G3_SHEFR, Q4FTU5_PSYA2, and Q3KFC3_PSEPF) is indicative of the stability of the conformation. The representative protein for state 4 is notably B9KYG0_THERP, the outstanding thermophile mentioned above, whose “tail” domain is bonded to the region at the back of the active site domains resulting in a higher packing score than other thermophiles. It was posited that this increase in residue interaction within the interior of the protein aids in stability at higher temperatures (B9KYG0_THERP was simulated at its observed 346 K) by anchoring the internal structures associated with the catalytic residues.

### 3.4. Salt Bridges, but Not Disulfide Bonds, Stabilize Thermophilic Proteases

While protein stability can be enhanced by subtle factors such as cohesion, it can also be more directly ensured by the use of strongly bound interactions, such as salt bridges and disulfide bonds. In this set of proteins, the only disulfide bonds identified are within the S13s, all of which are psychrophilic or mesophilic; cysteine is, in general, relatively exiguous within the set, and we observe no cysteine pairs within bonding distance among the thermophiles. Disulfide bonding is thus not a thermally stabilizing adaptation for these proteins.

Looking to other bound interactions, we do not see higher numbers of hydrogen bonds among thermophilic enzymes (see Appendix A). Salt bridges, however, follow a somewhat different pattern. Salt bridges were found using the saltbridge extension in VMD for all sequences, and the results are plotted in Figure 15. Thermophiles are observed to have a higher number of salt bridges in their structure than their psychrophilic homologs. This is in accordance with many other studies of stability of thermophiles, a summary of which can be found in Vieille and Zeikus [70].

## 4. Discussion

### 4.1. For Prediction of Seed Structures for Novel Proteins, Comparative Modeling Can Outperform Deep Learning

Our process for generating molecular models depends on the relaxation of the starting structure using molecular dynamics after modifying the protein in accordance with other known information about its maturation, e.g., removal of signal sequences and addition of disulfide bonds. Generating accurate models, therefore, depends on having the most realistic structure prediction available. Here we investigated both iTasser, which uses comparative modeling based on experimentally determined template structures, and AlphaFold, a more recently developed program that uses deep learning (Figure 16). In prior work, the comparative modeling approach has been successful at predicting structures for soluble enzymes, such as those studied here (e.g., [50,51,71,72,73]), but AlphaFold has gained considerable attention for its remarkable performance in the CASP competition [36] (often regarded as a definitive indicator of overall prediction quality). We were thus interested to see whether this novel approach would yield comparable performance on our enzyme set. We found that for the sequences of interest, iTasser generated acceptable (i.e., plausible and well-folded) starting structures for 32/32, whereas AlphaFold did for only 5/32. The goal of our procedure is always to produce physically reasonable models that can be used as a starting point for the MD simulations and that agree with the biological and structural information that is known.

In almost all cases, the AlphaFold structures are characterized by a highly extended unfolded region at the N-terminus, which is shown in the left panels of Figure 16A,B, as well as many more examples in Appendix A. In principle, this could represent an unstructured region; however, all but three of these proteins (A0A0H3JIB3_ECO57, A0A106BYZ8_SHEFR, Q086W1_SHEFN) are predicted to have N-terminal signal sequences, which are generally helical [75,76]. In the left panel of Figure 16A, the residues corresponding to the signal sequences are highlighted in yellow on both structures, showing that they are in the expected helical conformation for the iTasser model and in an unrealistic extended conformation in the AlphaFold model. Further, in the cases where the structures of homologous proteins are known, other structural details are captured more closely by the iTasser structures than by AlphaFold. Figure 16A(i,ii) shows the crystal structure of Penicillin-Binding Protein 5 (PBP5) from *Pseudomonas aeruginosa* (PDB ID: 4K91) [74] overlaid with the iTasser and AlphaFold predictions, respectively. Here, the C-terminal domain is highlighted in yellow, illustrating that its conformation is much closer to that of the experimentally determined structure for iTasser than for AlphaFold. Figure 16B shows another example where the N-terminal signal sequence is correctly folded into a helix in the iTasser model and unfolded in the AlphaFold model, along with a longer N-terminal sequence that is folded against the catalytic domain in the iTasser structure. Figure 16B(i,ii) shows one part of this protein overlaid with the crystal structure of DACC_BACSU (PDBID: 2J9P [68]), which is also an S13 protease. The sequence region shown in yellow is compactly folded as part of the small β-domain in the iTasser structure and in an unlikely extended conformation in the AlphaFold structure. Finally, in the case where there is an experimental structure of that specific protein (again DACC_BACSU, which is shown in Figure 16C), both structure predictors perform very well overall, as expected. Here, iTasser predicts a not entirely helical conformation for the signal sequence (yellow), while AlphaFold predicts an unbroken helix that is 46 residues long, much longer than the typical signal sequence length of 15–30 residues [77] and the 29 residues predicted by SignalP 6 (Appendix A) [29]. However, both predictors agree very well with the experimental structure for the catalytic domain (Figure 16C(i,ii)).

Overall, we observe that structural predictions from AlphaFold were extremely poor for this protein set, with sufficiently serious deficiencies to be immediately apparent upon even cursory examination. By contrast, iTasser reliably produced structures that were well-folded and compatible with the available evidence. This difference is strikingly at odds with CASP outcomes, suggesting that the competition may not always be a good indicator of performance for comparative studies, such as that undertaken here. While it may be tempting to conclude that more recent, deep learning-based prediction schemes are necessarily superior to older methods, our experience shows that this is not universally the case. We caution against using predicted structures without first inspecting them for obvious problems and suggest care when adopting prediction methods with a relatively limited history of practical use.

### 4.2. Conventional Wisdom on Protein Adaptation to Extreme Thermal Environments May Not Generalize to All Enzyme Classes

It is tempting to generalize the adaptations found in homologous proteins from different extreme environments and assume most proteins follow similar patterns of adaptation. At the most basic level, it seems intuitive that fitness-increasing mutations made in response to thermodynamic variables in the environment should be similarly selected for across proteomes. This assumption can be tested, in part, by sequence analysis. In this study, sequence clustering did partially split the sequences by thermal groups, dividing mesophiles between them. Psychrophiles were then further split based on the presence of a tail domain believed to assist with localization to the cell wall. Without knowing additional information about the environment and the structures of the proteins, the results of the sequence clustering could be interpreted as evidence that proteins adapt to different thermal environments by simple, compositional changes that hold across enzymes. Our compositional analysis, however, suggests a more nuanced view: the two psychrophilic clusters often vary as much or more from each other as from the thermophiles and, in some cases, in opposite directions. Further, as shown in Table 1, a number of the trends in the variation of amino acid concentration with temperature seen here differ from those reported in studies of other systems [70,78]. We observe that these prior studies have not themselves reported entirely consistent results, implying that temperature/composition relationships may depend more upon the properties of specific protein families than on general, thermodynamic considerations.

The structural stability of a protein, as it relates to temperature, is frequently associated with some measure of packing, with an underlying assumption that a more tightly packed protein is inherently more thermodynamically stable. In the context of thermal adaptation, this raises the question: what adaptations can a protein make to increase packing when an increase in stability is needed, such as in the presumably high-energy systems of thermophiles? It is widely argued that high-temperature proteins leverage the use of the hydrophobic effect for stabilization, following the logic that an increase in the size and/or number of hydrophobic residues will allow proteins existing at higher temperatures to take advantage of the hydrophobic effect to achieve a tighter, and, therefore, more stable core. Despite its persistence, the claim that stability at high temperatures should correlate with a dense hydrophobic core does not seem to be well-substantiated, and studies to support such a claim is scant. Other measurements for packing density have shown little difference between the packing of proteins found in mesophiles and thermophiles [79,80] and an increase in the packing density of the proteins in psychrophiles [81], contradicting the assumed relationship between packing and stability. The current study also finds a trend of decreasing packing density with increasing temperature, with little noticeable difference between thermophiles and mesophiles. Additionally, this study finds a decrease in cohesion with increasing temperature across all levels of structure, suggesting a more flexible structure in the thermophiles of this set. These observations contradict the notion that structural stability at high temperatures is dependent upon protein rigidity and density, and instead suggests that other stabilizing mechanisms are in use by D-Ala-D-Ala carboxypeptidases.

Our observations on the role of stronger interactions on protein stability [82,83] are more mixed. While disulfide bonds can play this role in some systems [62], they do not do so here (Indeed, we see disulfide bonds here only among enzymes from psychophiles and mesophiles). On the other hand, we do see a clear enhancement in the incidence of salt bridges within the thermophilic enzymes in our set, following the commonly proposed trend within the literature.

Clearly, there are numerous mechanisms by which a protein may maintain stability and function in harsh environments, and our observations align with the view that different mechanisms may be employed in different cases. As such, broad generalizations regarding the relationship of amino acid composition, packing, local cohesion, prevalence of disulfide bonds and/or salt bridges, etc., may be of limited use when studying specific protein families. Alternately, it is plausible that there are other, as yet unidentified, thermal adaptations that allow particular enzymes to maintain function under extreme conditions. More detailed mechanistic studies of enzyme function across temperature regimes will undoubtedly shed light on this issue. At the same time, it continues to be useful to use broad, sequence-based and computational studies to help determine what patterns of adaptation do persist and to identify when and where they are likely to be held.

**Table 1 biomolecules-13-00328-t001:** Selection of Literature Claims Regarding Protein Structure vs. Temperature.

Claim References	Examined Proteins	Results	Comparison with Current Study
Number and/or size of specific amino acid types			
Kannan and Vishveshwara [84]	24 meso and thermo. homologues	increase in aromatic networks/clusters w/ increased Temp.	disagrees (similar amounts of aromatic residues)
Vieille and Zeikus [70]	8 mesophilic 7 hyperthermophilic organisms	increase E, G, I, K, P, R, V, W, Y w/ increased Temp.	disagrees (G, I, R, and W split, more P in psychro)
Kumar et al. [78]	6 each psychro- meso- and thermophilic β-D-galactosidases	more A, G, S, R in psychrophiles, more V, Q, E, F, T, Y in thermophiles	disagrees (A, G, and R split, more Q, F, T, in psychrophiles)
Density/packing and Rigidity/Flexibility			
Karshikoff and Ladenstein [79]	80 and 24 proteins from meso and thermo organisms, respectively	packing density is similar between meso and thermo	agrees
Radestock and Gohlke [85]	19 homologs protein pairs from meso and thermo organisms	increased rigidity in thermophiles	disagrees (more unstructured, lower cohesion)
Wells et al. [86]	citrate synthase	increased rigidity in thermophiles	disagrees (more unstructured, lower cohesion)
Amadei et al. [81]	57 thermophilic and mesophilic pairs	decreased density with increased temperature	agrees
Sen and Sarkar [80]	17 homolog thermo-meso pairs, 18 homolog psychro-meso pairs	no difference in average packing factor	disagrees (increased packing trend for psychro)
Number of disulfide bonds			
Appleby et al. [62]	5${}^{\prime }$-deoxy-5${}^{\prime }$-methylthioadenosine phosphorylase *Solfolobus solfataricus*	disulfide bonds increase thermal stability	disagrees (too few Cys found for disulfide bonds to form)
Electrostatic interactions and/or salt bridges			
Szilágyi and Závodszky [82]	64 meso and 29 thermo homologs	increase in ion pairs w/ increased growth Temp.	agrees
D’Amico et al. [87]	psychrophilic α-amylase	decreased weak interactions in psychrophiles	agrees
Chan et al. [83]	thermophilic ribosomal protein L30e	increase in salt-bridges stabilize thermophiles	agrees
Niu et al. [88]	1,3-1,4-β-glucanase	increased stability at high Temp. with K→S substitutions	disagrees (higher K conc. at high temp.)

## 5. Conclusions

Comparative modeling of S11, S12, and S13 proteases from multiple organisms suggests a number of patterns that may be useful targets for further work. Among the notable findings are the following.

Our examination of packing as a function of environmental temperature across the proteases studied here suggests a clear pattern, with mean RSAs increasing with temperature (i.e., psychrophilic proteases have more buried residue surface area than thermophilic proteases, on average). For the proteases studied here, this pattern seems to be related not merely to local packing properties but to the broader fold (including the presence or absence of extended structures with a high surface area). An empirical evaluation of this pattern is of obvious interest, as is the question of how broadly it applies among serine hydrolases; the relationship between native temperature and higher-order (i.e., domain-level) structure is, in particular, poorly understood, and further comparative analysis would help to determine whether trends such as those seen here are representative versus idiosyncratic.

Surface dynamics is another area in which we see trends that involve both local structure and higher-order features (e.g., the presence and behavior of the tail domain), with distinctive patterns for thermophilic versus psychophilic proteases. These solvent interaction patterns are natural targets for NMR studies (possibly in conjunction with EPR) and may provide clues regarding both protein stability adaptations and potential substrate interactions.

At a more basic level, we also observe interesting differences in tertiary structure (particularly, the role of the “tail” domain) across the proteases studied here. B9KYG0_THERP emerges as a notably interesting target for further investigation, as we predict it to have a very different tail domain structure from that of the other thermophilic enzymes studied here. Better characterization of this protein may help elucidate the role of the tail domain within these classes more broadly.

Finally, we observe that detailed conformational analysis of the catalytic residues performed here shows patterns that suggest relatively large differences across proteins and that do not always conform to a “typical” serine hydrolase pattern. Our analysis of static structures is compatible with the hypothesis that the three-residue MEROPS model of catalysis is incorrect and that a four-residue model is needed for these proteins. Alternately, substrate binding may play a critical role in reshaping the active site for some proteins in the set. Both would seem to be promising directions for further study.

In conclusion, we note that continuing advances in structure prediction, efficient molecular dynamics simulation, and exploratory data analysis for high-dimensional data are greatly increasing our ability to “scout” for patterns in protein sets that would be too large or otherwise difficult to approach experimentally. While such studies are always tentative in character, they give us a useful lens with which to focus our experimental investigations and through which to gain greater insights into the molecular basis of biological adaptation.

## Figures and Tables

**Figure 1 biomolecules-13-00328-f001:**
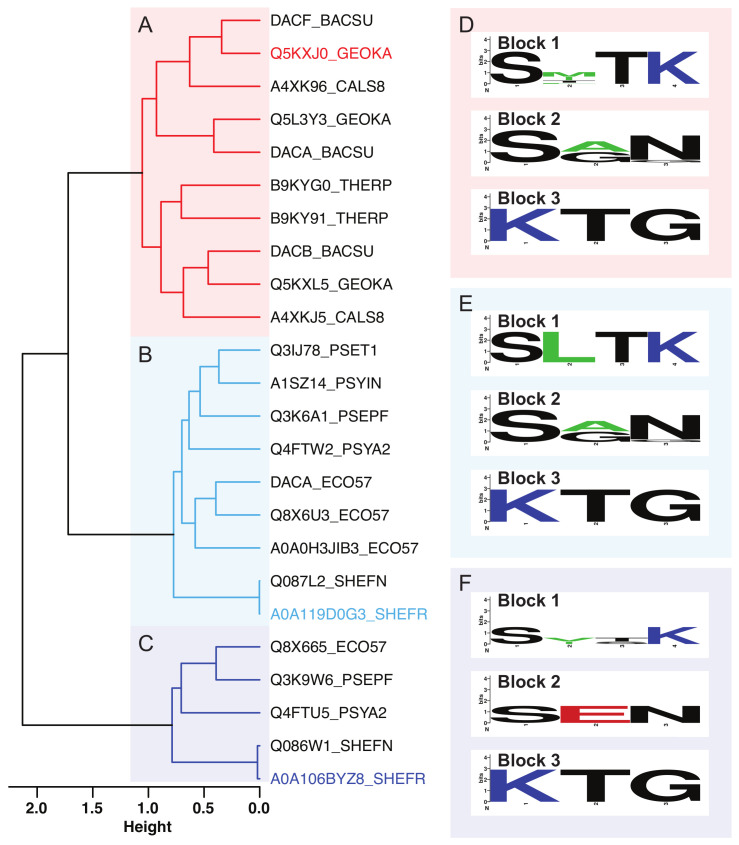
Microbial S11 proteases clustered according to protein sequence similarity. The sequences are described by three clusters. The thermophilic and psychrophilic sequences are separated, whereas mesophilic sequences are distributed among the three. Cluster (**A**) contains thermophilic sequences as well as those from *B. subtilis*. Clusters (**B**,**C**) comprise proteins from psychrophilic organisms, *P. fluorescens* and *E. coli*. The highlighted sequence name in each cluster indicates the centroid in sequence space. Panels (**D**–**F**) contain sequence logos associated with the three conserved active residue sequence blocks for each protein cluster.

**Figure 2 biomolecules-13-00328-f002:**
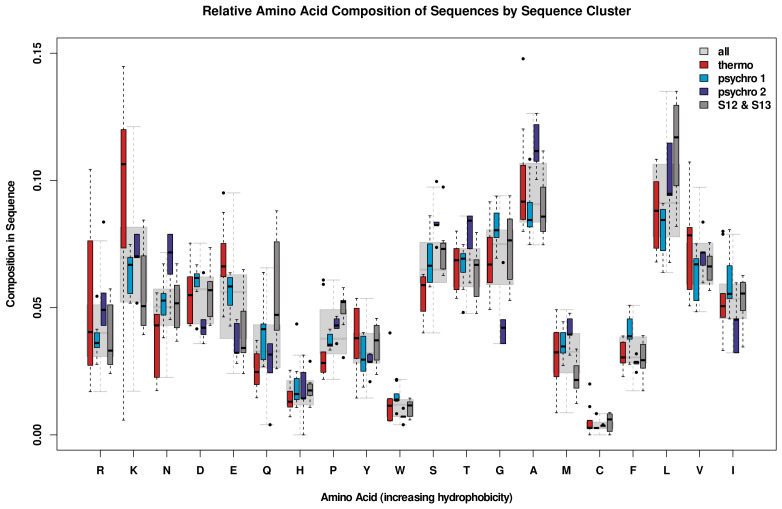
Amino acid compositions normalized by sequence length and by thermal group.

**Figure 3 biomolecules-13-00328-f003:**
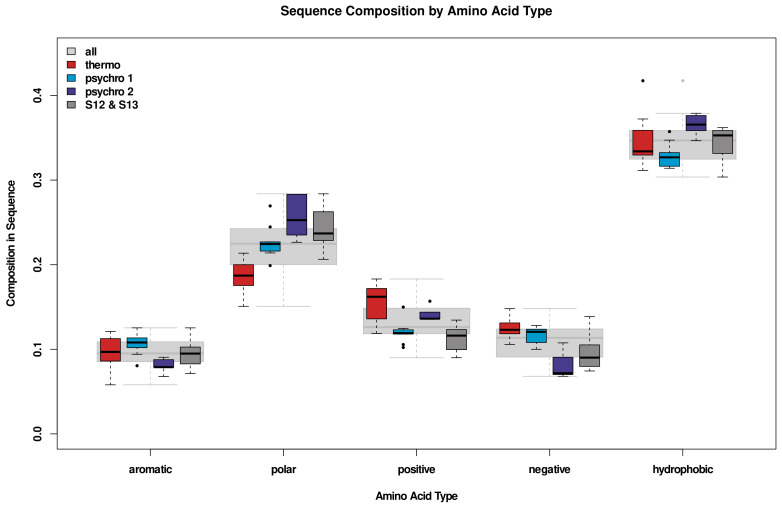
Differences in overall composition by amino acid type. Thermophiles are relatively exiguous in uncharged polar amino acids and enriched in charged residues. On the other hand, psychrophiles are relatively enriched in uncharged polar residues; however, their composition differs considerably by sequence cluster.

**Figure 4 biomolecules-13-00328-f004:**
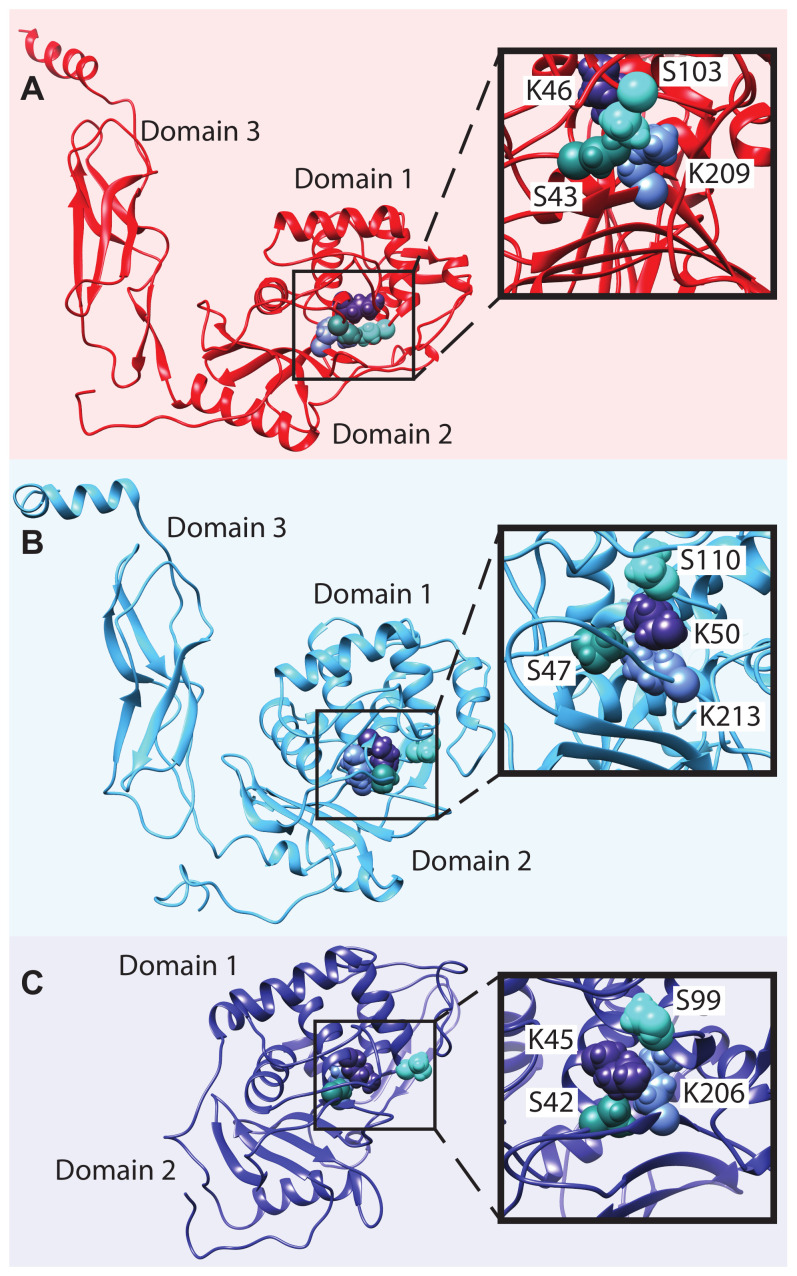
Representative structures of bacterial S11 proteases. The structures chosen are for the proteases representing the centroids of the three clusters shown in Figure 1. (**A**) Q5KXJ0_GEOKA, (**B**) A0A119D0G3_SHEFR, and (**C**) A0A106BYZ8_SHEFR. The insets of each panel show the active site residues, including the catalytic SKS triad (labeled in dark cyan, dark blue, and light cyan, respectively), as well as the additional K believed to be important in substrate binding and stabilization of the transition state (light blue).

**Figure 5 biomolecules-13-00328-f005:**
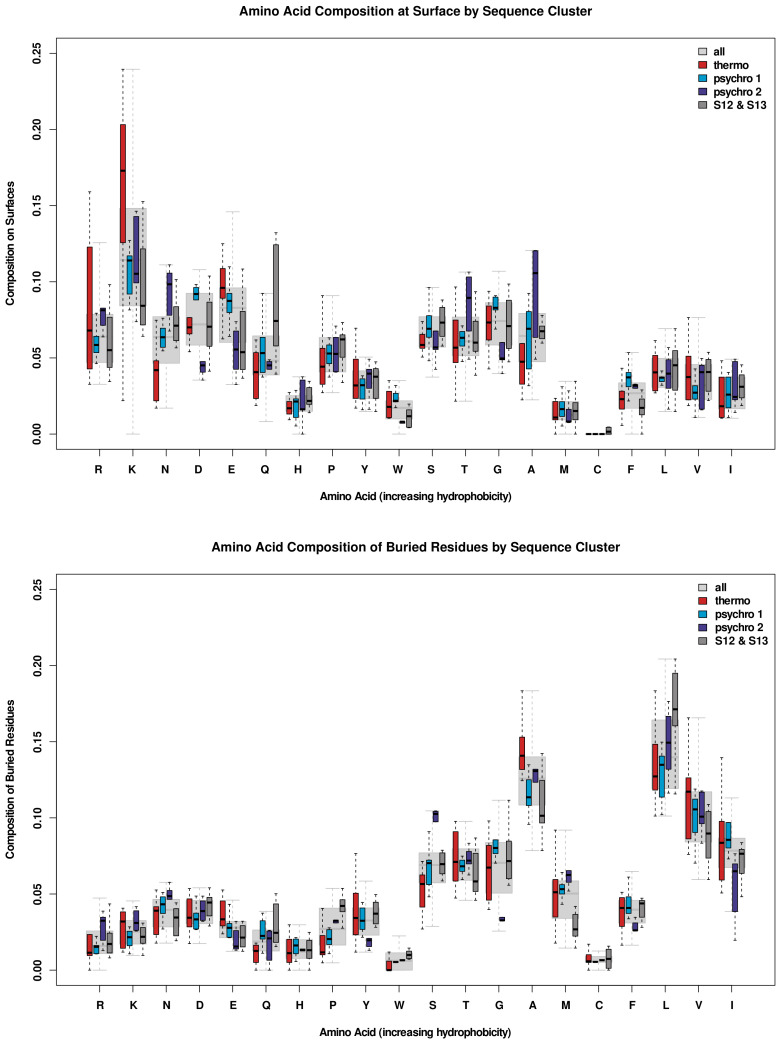
Rate of exposure (**top**) and burial (**bottom**) of residues by amino acid type for all proteases in each sequence cluster (Figure 1), as well as for S12 and S13 sequences (combined). “Exposed” residues had relative SASA values greater than 0.2, with the remaining residues considered to be buried.

**Figure 6 biomolecules-13-00328-f006:**
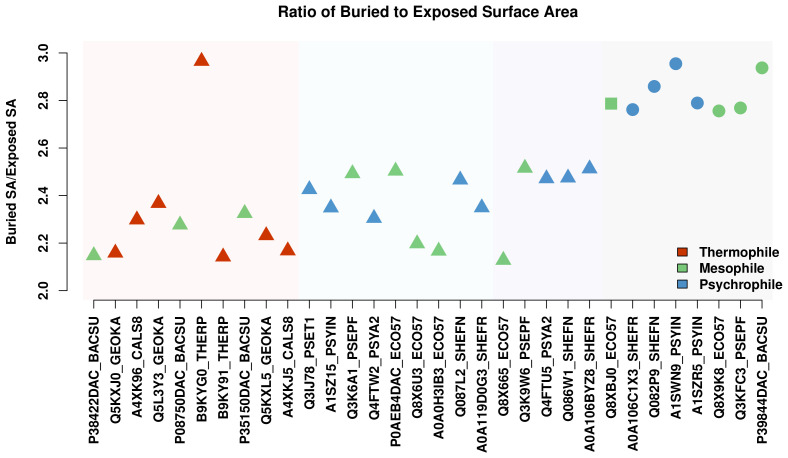
Ratio of external to internal surface area as a relative measure of packing, arranged by sequence similarity. In general, a greater fraction of residue surface area is buried in low-temperature proteases; a notable outlier, B9KYG0_THERP, is discussed below.

**Figure 7 biomolecules-13-00328-f007:**
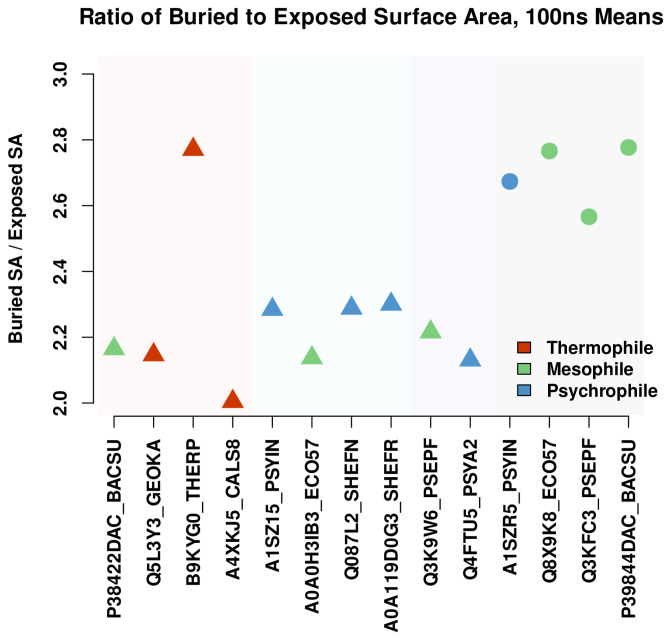
Ratio of external to internal surface area as a relative measure of packing for 100 ns simulations, arranged by sequence similarity. Trajectory means were determined using parametric bootstrap confidence intervals for accounting for autocorrelation over 5000 frames; 95% confidence intervals are smaller than plotted points and hence not visible. The results are consistent with those of Figure 6.

**Figure 8 biomolecules-13-00328-f008:**
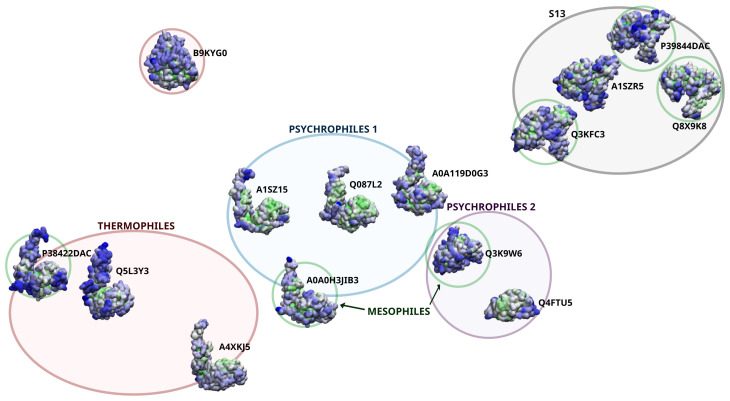
Dynamic solvation index values (*D*), by protein; layout based on Figure 7. *D* value coloring ranges from green (low) to blue (high). High *D* values are evident in the tail domain, with psychrophiles showing particularly low values of *D* on select regions opposite the active site.

**Figure 9 biomolecules-13-00328-f009:**
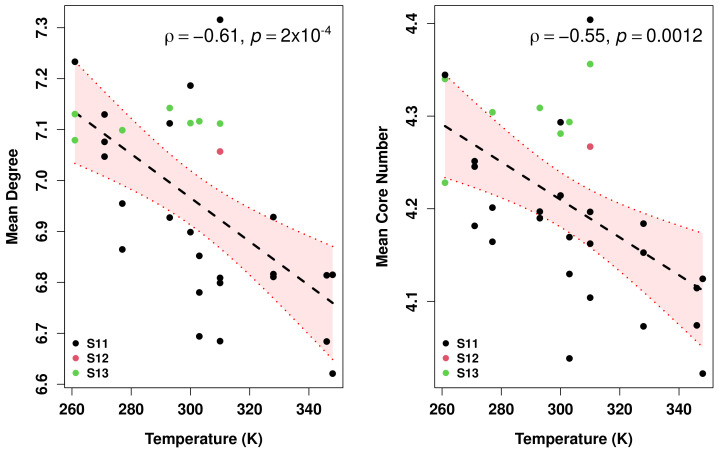
PSN cohesion by temperature, as assessed by mean degree (**left**) and mean core number (**right**). OLS fit shown by central line; the shaded area indicates the 95% confidence bands. For both measures, we see a significant decline in cohesion for proteins with higher observed environmental temperatures.

**Figure 10 biomolecules-13-00328-f010:**
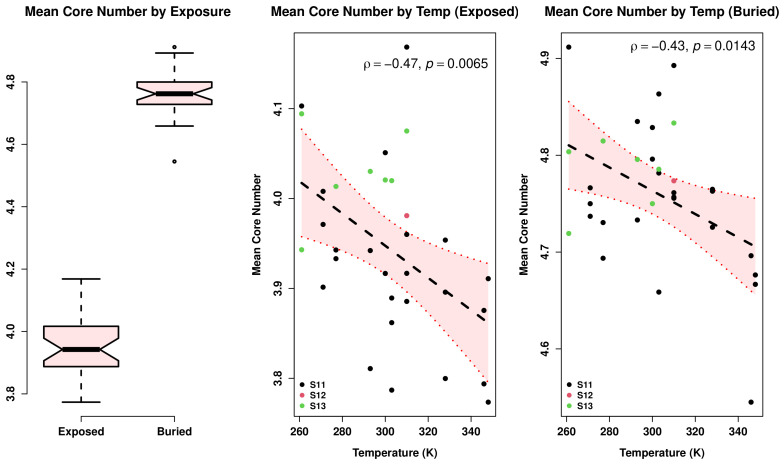
(**left**) Mean core number for moieties in exposed versus buried residues. (**right**) Relationship between mean core numbers and observed environmental temperature, considering only exposed or buried residues, respectively (OLS fits shown by central lines; shaded area indicates the 95% confidence bands). While buried groups are generally in much more cohesive positions, environmental temperature has a similar association with cohesion for exposed and buried groups.

**Figure 11 biomolecules-13-00328-f011:**
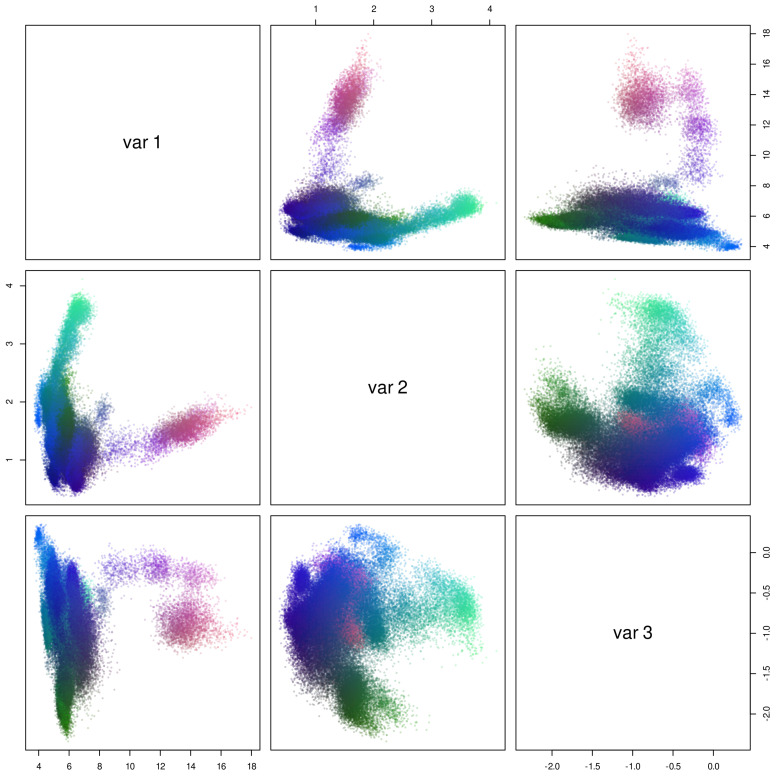
Deep random feature embedding of conformations of the catalytic (SKS) heavy atoms over all trajectories (three highest–variance dimensions). Coloring corresponds to the position on all three dimensions in RGB space (red = first, green = second, blue = third).

**Figure 12 biomolecules-13-00328-f012:**
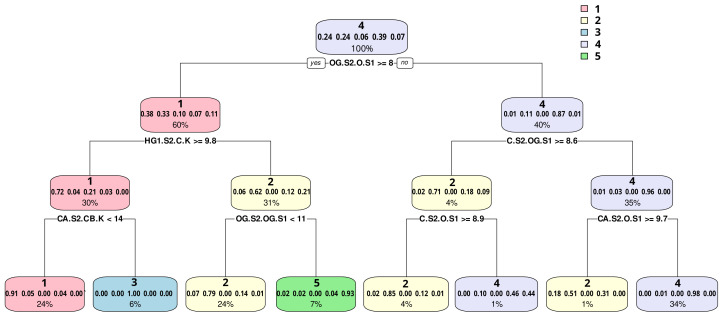
Classification tree to separate conformational clusters using interatomic distances. Nodes list (in rows) the dominant cluster for points beneath the node, a fraction of points in each cluster beneath the node, and the fraction of the data set beneath the node. A small number of key distances approximately characterize the cluster states.

**Figure 13 biomolecules-13-00328-f013:**
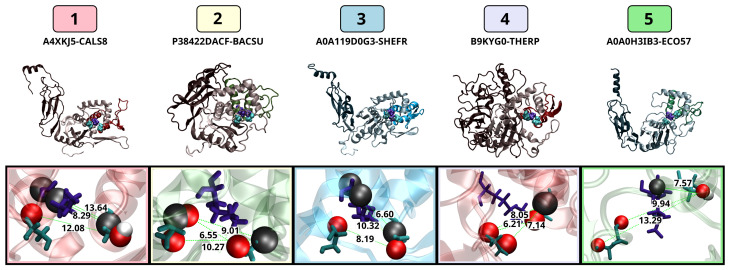
Most central conformations for each conformational state; catalytic SKS residues shown in atom/bond representation, with larger spheres highlighting atoms selected by classification tree (Figure 12) to distinguish conformations. Distances noted are in Angstroms and were calculated in VMD.

**Figure 14 biomolecules-13-00328-f014:**
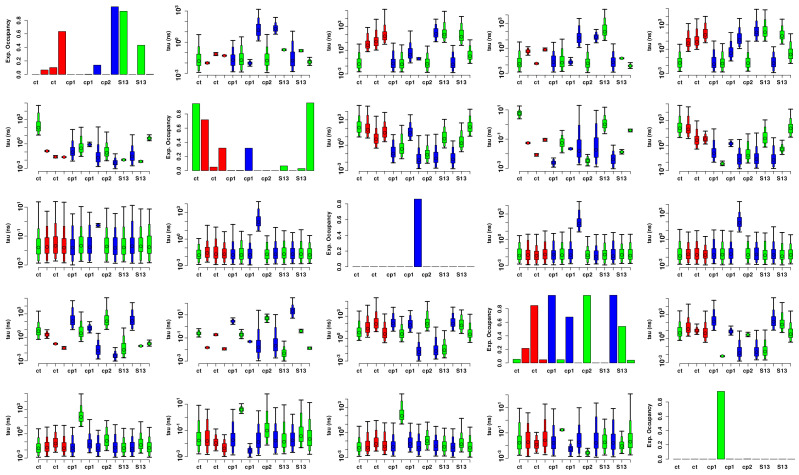
Posterior transition rate and occupancy estimates from the Markov model. States indicated by row/column. Diagonal panels show posterior occupancy probability by protein; colors indicate the thermal group. i,j off-diagonal panels show 95% posterior intervals for estimated waiting times for transitions from state *i* to state *j* (in ns), by protein (Note that uncertainty is high for transitions involving states not observed for a given protein, leading to wide posterior intervals).

**Figure 15 biomolecules-13-00328-f015:**
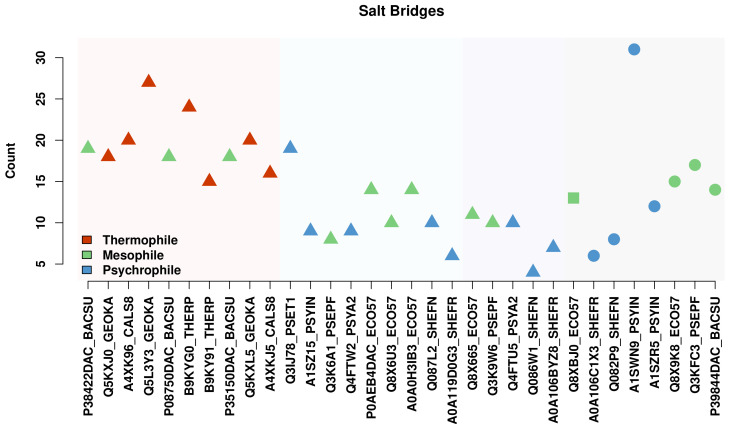
Counts of salt bridges by protein. Thermophiles show higher salt bridge counts overall, though two psychrophiles show comparable numbers.

**Figure 16 biomolecules-13-00328-f016:**
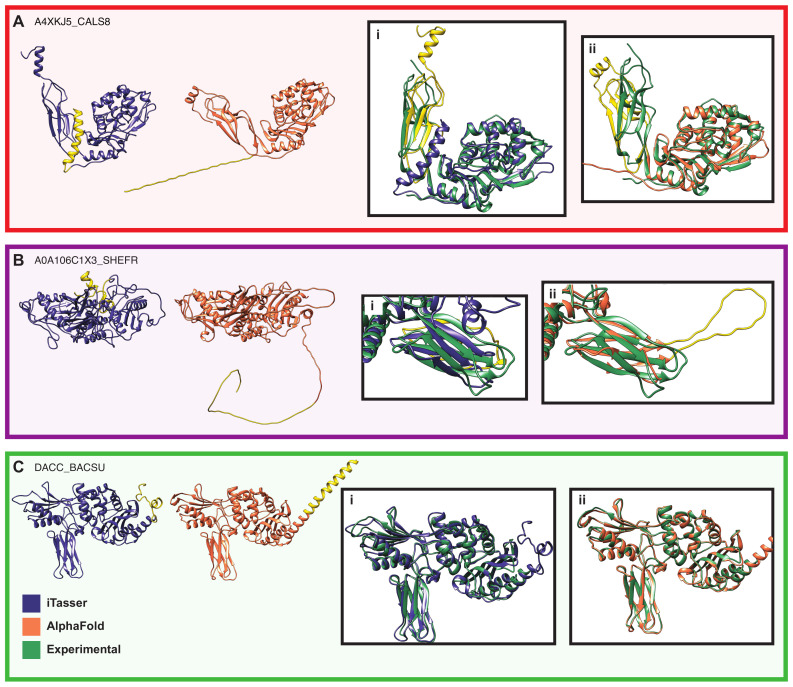
Comparisons between models produced by iTasser and AlphaFold. (**A**) The iTasser (blue) and AlphaFold (coral) models for A4XKJ5_CALS8 have similar folds for the catalytic domain and the C-terminal “tail” domain. However, the N-terminal signal sequence (yellow) is in the expected helical conformation in the iTasser structure and an unusual extended state in the AlphaFold structure. The insets i. and ii. show these structures overlaid with the crystal structure of a homolog (PDB ID: 4K91 [74]). The iTasser structure does better at reproducing the relative orientation of the two domains. (**B**) The iTasser (blue) and AlphaFold (coral) models for A0A106C1X3_SHEFR behave similarly to the ones in (**A**), but here the unfolded N-terminal region extends far beyond the signal sequence. The insets i. and ii. show a compact β-sheet region overlaid with the experimental structure of DACC_BACSU (see (**C**)), which is also an S13 protease. In the iTasser structure, this small domain is compactly folded, while the AlphaFold structure predicts an unusual extended loop. (**C**) The iTasser (blue) and AlphaFold (coral) models for DACC_BACSU are both well-folded and compact. Neither exactly produces the expected α-helix for the signal sequence; in the iTasser structure, it is not fully helical, and in the AlphaFold model, the helix is much longer than is typical. Both models (i. and ii.) are in excellent overall agreement with an experimental crystal structure of this protein (green, PDBID: 2J9P [68]). The signal sequence is not present in the construct that was experimentally solved.

## Data Availability

Data is available in the Appendix A and upon request.

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
