# Peer review of "Comparative Modeling and Analysis of Extremophilic D-Ala-D-Ala Carboxypeptidases"

_biomolecules, 2023, doi:10.3390/biom13020328_

Round 1

Reviewer 1 Report

Dear authors, after a careful revision of the manuscript entitled "Comparative Modeling and Analysis of Extremophilic D-Ala-D-Ala Carboxypeptidases" describing some in silico analyses (Protein Structure Prediction- I-Tasser, Molecular Dynamics) I judged it able to be published in this journal.

Author Response

We thank the reviewer for their consideration.

Reviewer 2 Report

The authors constructed  comparative models for over 30  proteases from S11, S12, and S13 subfamilies. The manuscript describes also basic structural properties of these structural models.

Major comments

1) In my opinion, the manuscript lacks its main goal. I found a wide variety of computational results, but I'd also like to a see a main hypothesis they support. There should be a single, well pronounced biological question this data and these results support (or deny)

2) Results should be presented in more convincing way, e.g:

 - I'd expect energy vs time plots made from MD simulations. These are essential to convince the readers about the convergence of the simulations. rmsd measured to the starting model (as a function of MD time) should also be provided. It'd be lots of plots, so some of them might be moved to supporting materials

 - Surface area in the figures 3 and 4 should be presented with error bars rather than just averages over MD trajectories

 - It's unclear, how the selection of force field affects the results presented in this work It would be very valuable to confirm the results in Amber or Gromos force fields

Minor comments

1) Methodology used by the authors is quite obsolete. iTasser was a great tool 7 years ago, but it has been surpassed by the new generation of deep learning approaches. Authors should at least make the models with AlphaFold2 / RosettaFold / OpenFold and compare results with the models they already have.

Author Response

We thank the reviewer for their thoughtful remarks, and have edited the paper in response. Below we include a summary of changes; we hope that all parties will agree that they have improved the paper.

  • Reviewer 2: In my opinion, the manuscript lacks its main goal. I found a wide variety of computational results, but I'd also like to a see a main hypothesis they support. There should be a single, well pronounced biological question this data and these results support (or deny)

    • We have reorganized the manuscript around its main goal of providing a holistic understanding of the adaptive mechanisms employed by D-Ala-D-Ala carboxypeptidases in extreme thermal environments. In doing so, we have also systematically compared our results to common trends promoted in the literature, and we include a discussion of the generalizability of such trends across proteins regardless of function or environment.

  • Reviewer 2: I'd expect energy vs time plots made from MD simulations. These are essential to convince the readers about the convergence of the simulations. rmsd measured to the starting model (as a function of MD time) should also be provided. It'd be lots of plots, so some of them might be moved to supporting materials

    • Energy plots and RMSD plots have been provided and can be found in the supplemental information.

  • Reviewer 2: Surface area in the figures 3 and 4 should be presented with error bars rather than just averages over MD trajectories

    • As noted in the main text, the 95% confidence intervals for the trajectory means (previously Figure 4, now Figure 7) are present, but they are too small to be visible on a plot (given that they are taken over 5000 frames of each trajectory). The results obtained from equilibrated snapshots (previously Figure 3, now Figure 6) are for single structures of each sequence (not sample means), and so standard error calculations are not applicable.

  • Reviewer 2: It's unclear, how the selection of force field affects the results presented in this work It would be very valuable to confirm the results in Amber or Gromos force fields

    • We agree that examining the effects of one’s choice in force field would be an interesting study; however, that is beyond the scope of the current paper. We observe that the force field used here (CHARMM36m) is a well-validated and widely used force field for modeling of globular proteins, and we are not aware of any specific results suggesting deficiencies that would alter the results of this study. We also note that a comparison of the sort described would require replicating the entire ensemble of simulations and analyses performed in the paper multiple times, and the turn-around time given by the publishers for revisions was too short for running the large number of simulations that would be required.

  • Reviewer 2: Methodology used by the authors is quite obsolete. iTasser was a great tool 7 years ago, but it has been surpassed by the new generation of deep learning approaches. Authors should at least make the models with AlphaFold2 / RosettaFold / OpenFold and compare results with the models they already have.

    • It is true that deep learning approaches have recently become popular in this area, and have done well in the CASP competition. However, their actual performance on real-world systems often falls short of their performance in trials. Specifically, we did examine AlphaFold2 as a structure predictor, and found that it consistently produced predictions that were of markedly poor quality (in contrast to iTasser, which did not). Since this is a topic of evident interest, we have provided a detailed examination of the differences between AlphaFold and iTasser predictions (and comparisons with solved structures, where available), showing how AlphaFold consistently fails to produce realistic structure predictions for this protein set (while iTasser, by contrast, performs very well). We believe that this points to a general weakness of the CASP competition (i.e., performance in that competition is not necessarily predictive of real-world performance), and hope that it provides guidance to others in the field - "classic" methods can turn out, at least in some cases, to outperform more recently developed techniques.

Reviewer 3 Report

This paper is a totally computer-driven analysis of specific bacterial cell wall enzymes from organisms with different growth temperature requirements. The major message is not clearly presented, and to help with this, the paper could stand more text in the Conclusions section that would give biochemists and microbiologists who are not structural biologists something more to take from the results. In general the paper is well written, and so I have just a few specific comments.

Specific comments:

1. The second paragraph of the M&M should be converted into an informative table that would include, in addition to the name of the organism and the database references, the following: (1). Habitat of the organism; (2) Growth temperature optimum of the organism; (3) Gram-stain reaction of the organism; (4) Citation to the description of the organism. Such data should be easily retrievable with a little digging into the literature back to the paper that describes the organism as a new species or genus/species. The problem with the presentation as it stands is that the labels "thermophile", "psychrophile", and "mesophile" are moving targets and hardly precise. A microbiologist reading this paper might wonder "Does this thermophile grow optimally at 80oC or 50oC?" Both would be considered "thermophiles". The authors are lumping some perhaps quite different physiologies into broad categories and more information on the organisms will better inform the reader about the subject matter and perhaps may even bring additional insight into the structures the authors are interested in.

2. The authors often use the phrase "thermophilic/psychrophilic/mesophlic sequences" (e.g. in the legend to Fig. 1 or on line 172). The sequences themselves do not have temperature relationships, only the organisms. Please rephrase.

3. Line 215 et seq and Fig. 3: It has indeed been thought for some time (and with evidence to support it) that enzymes from thermophilic bacteria have (and logically should have) more tightly packed folding, with psychrophiles having just the opposite, more flexible foldings. Surprisingly, the authors' findings were in opposition to this. This is of interest; however, the authors need to better acknowledge previous work by bringing in more literature from work that has shown tighter packing in enzymes from thermophiles and do a better job of rationalizing their results in light of previous research to the contrary. Also, do their findings in this regard give the authors any concerns that their methods or the assumptions they are using may not be valid?

4. Fig. 3. To this reviewer with poor color vision, these symbols all look about the same. And the black circle in the middle of each symbol simply makes matters worse. Please improve color presentation in this figure.

5. Line 233: the word "Fig." needs to be inserted before the number "3".

6. Line 281, 359, and perhaps elsewhere: express all temperatures in degrees Celsius, not in degrees Kelvin. No need to make the reader run to a temperature conversion chart to get a meaningful number. Even biochemists tend to think in degrees Celsius.

7. Line 311: What is meant by the phrase "highest-dimensional dimensions"? 

8. Lines 378/379. This is not a complete sentence.

9. Conclusions: As mentioned earlier, the greater the "take-home" lesson you can provide to the non-expert, the more science they will take away from your paper and the more interest the paper will generate. Perhaps this would be a good time to explain the very surprising results that were found concerning buried vs. exposed surface area in the proteases plotted in Fig. 3. Lacking such a firm rationalization, I believe many readers might question your methods and assumptions, and in so doing, not really appreciate the results.

Author Response

We thank the reviewer for their thoughtful remarks, and have edited the paper in response. Below we include a summary of changes; we hope that all parties will agree that they have improved the paper.

  • Reviewer 3: The second paragraph of the M&M should be converted into an informative table that would include, in addition to the name of the organism and the database references, the following: (1). Habitat of the organism; (2) Growth temperature optimum of the organism; (3) Gram-stain reaction of the organism; (4) Citation to the description of the organism. Such data should be easily retrievable with a little digging into the literature back to the paper that describes the organism as a new species or genus/species. The problem with the presentation as it stands is that the labels "thermophile", "psychrophile", and "mesophile" are moving targets and hardly precise. A microbiologist reading this paper might wonder "Does this thermophile grow optimally at 80oC or 50oC?" Both would be considered "thermophiles". The authors are lumping some perhaps quite different physiologies into broad categories and more information on the organisms will better inform the reader about the subject matter and perhaps may even bring additional insight into the structures the authors are interested in.

    • We agree that consolidating this information into an accessible table is useful for readers, and have provided a table containing the organism name and database references, habitat where the selected strain was collected, the observed environmental temperature recorded at the time of collection, gram-stain information, and citations to description of the organism (Supplementary Tables 2 and 3). We are unable to include the optimal growth temperatures, since there are many complications involved with defining that temperature for each organism (and in many cases, no clear experimental determination has been made). However, we do report the observed environmental temperature from which the isolate sequenced for this study was found (this being the most biologically relevant temperature for understanding the specific variants of the enzyme obtained). The observed temperature is also the temperature at which each respective sequence was modeled, and so is consistently used throughout the study. Additionally, the “thermophile”, “mesophile”, “psychrophile” labels used in this study are the same as those found in the literature for these organisms, as cited in the supplementary tables. We have provided further details for our reasoning in the main text.

  • Reviewer 3: The authors often use the phrase "thermophilic/psychrophilic/mesophlic sequences" (e.g. in the legend to Fig. 1 or on line 172). The sequences themselves do not have temperature relationships, only the organisms. Please rephrase.

    • This shortened phrasing is commonly used throughout the literature on adaptations of enzymes from thermophilic/psychrophilic/mesophilic organisms, and provides an easier flow for the reader without creating any additional confusion. We have endeavored to ensure that, where present, our usage is clear from context and consistent with the literature.

  • Reviewer 3: Line 215 et seq and Fig. 3: It has indeed been thought for some time (and with evidence to support it) that enzymes from thermophilic bacteria have (and logically should have) more tightly packed folding, with psychrophiles having just the opposite, more flexible foldings. Surprisingly, the authors' findings were in opposition to this. This is of interest; however, the authors need to better acknowledge previous work by bringing in more literature from work that has shown tighter packing in enzymes from thermophiles and do a better job of rationalizing their results in light of previous research to the contrary. Also, do their findings in this regard give the authors any concerns that their methods or the assumptions they are using may not be valid?

    • We agree that it has been thought for some time that enzymes have more tightly packed folding, however the logic behind this conjecture is not as clear. Additionally, the empirical evidence for this conjecture is thin, and in some cases contradictory. To further explore how commonly held beliefs about protein structure may apply to this set of proteins, we have included additional analyses to provide a more holistic understanding of how D-Ala-D-Ala carboxypeptidases adapt to extreme environments, and have compared our results to multiple studies exploring similar hypotheses for other protein sets. A table outlining a selection of literature results in comparison with the results found in this study is provided in the main text (Table 1). We have found that many assumptions found in the literature do not generalize to this sample of proteins (and, quite possibly, not to others); they may indeed have been valid for the protein or set of proteins from which the hypothesis was formed, but do not seem to apply in all cases. We now summarize this at some length. In our view, identifying which generalizations do or do not hold up in different protein families is an important goal of comparative, multi-protein studies, and one contribution of this work is to provide evidence for greater complexity in the adaptation of proteases to different thermal environments than has previously been appreciated.

  • Reviewer 3: Fig. 3. To this reviewer with poor color vision, these symbols all look about the same. And the black circle in the middle of each symbol simply makes matters worse. Please improve color presentation in this figure.

    • Color presentation of this and similar figures has been adjusted.

  • Reviewer 3: Line 233: the word "Fig." needs to be inserted before the number "3".

    • Fig.” has been added before all figure reference numbers.

  • Reviewer 3: Line 281, 359, and perhaps elsewhere: express all temperatures in degrees Celsius, not in degrees Kelvin. No need to make the reader run to a temperature conversion chart to get a meaningful number. Even biochemists tend to think in degrees Celsius.

    • We provide a table in the supplementary information (Table 3) that provides the temperatures in Celsius. However, we note that conversion from Kelvin to Celsius can be done by subtracting the value 273.

  • Reviewer 3: Line 311: What is meant by the phrase "highest-dimensional dimensions"?

    • This has been rephrased to “the three dimensions with the greatest variation”.

  • Reviewer 3: Lines 378/379. This is not a complete sentence.

    • The word “as” has been removed to make a complete sentence.

  • Reviewer 3: Conclusions: As mentioned earlier, the greater the "take-home" lesson you can provide to the non-expert, the more science they will take away from your paper and the more interest the paper will generate. Perhaps this would be a good time to explain the very surprising results that were found concerning buried vs. exposed surface area in the proteases plotted in Fig. 3. Lacking such a firm rationalization, I believe many readers might question your methods and assumptions, and in so doing, not really appreciate the results.

    • We have restructured the paper to provide a more cohesive message on the adaptation of D-Ala-D-Ala carboxypeptidases, and outlined how the results relate to those in literature. The “take-home” lesson is thus: there are many known mechanisms by which a protein may adapt to a new environment, and individual mechanisms that are believed to be generalizable may not apply in every case, and certainly not in the case of D-Ala-D-Ala carboxypeptidases.

Round 2

Reviewer 2 Report

Authors have exhaustively addressed all my concerns. The changes they introduced in this revision substantially improved the manuscript. The manuscript got considerably longer but at the same time easier to follow. Now the conclusions seems supported by the evidence provided by the authors.